# Signaling Through Purinergic Receptor P2Y_2_ Enhances Macrophage IL-1β Production

**DOI:** 10.3390/ijms21134686

**Published:** 2020-06-30

**Authors:** Gonzalo de la Rosa, Ana I. Gómez, María C. Baños, Pablo Pelegrín

**Affiliations:** Unidad de Inflamación Molecular y Cirugía Experimental, Instituto Murciano de Investigación Biosanitaria IMIB-Arrixaca, Hospital Clínico Universitario Virgen de la Arrixaca, 30120 Murcia, Spain; gonzadelarosa@yahoo.es (G.d.l.R.); ana.gomez@ffis.es (A.I.G.); mcarmen.banos@ffis.es (M.C.B.)

**Keywords:** purinergic signaling, macrophages, inflammation, cytokines, extracellular nucleotides

## Abstract

The release of nucleotides during necrosis or apoptosis has been described to have both proinflammatory and anti-inflammatory effect on the surrounding cells. Here we describe how low concentrations of UTP and ATP applied during macrophage priming enhance IL-1β production when subsequently the NLRP3 inflammasome is activated in murine resident peritoneal macrophages. Deficiency or pharmacological inhibition of the purinergic receptor P2Y_2_ reverted the increase of IL-1β release induced by nucleotides. IL-1β increase was found dependent on the expression of *Il1b* gene and probably involving JNK activity. On the contrary, nucleotides decreased the production of a different proinflammatory cytokines such as TNF-α. These results suggest that nucleotides could shape the response of macrophages to obtain a unique proinflammatory signature that might be relevant in unrevealing specific inflammatory conditions.

## 1. Introduction

The control of the maturation and release of the proinflammatory cytokine interleukin (IL)-1β and IL-18 in macrophages by the NLRP3 inflammasome complex is a tightly controlled two-step process. An initial signal primes the production of the immature form of the cytokines and NLRP3. A second signal induces activation of the NLRP3 inflammasome complex with the subsequent activation of caspase-1 and the processing and release into the extracellular space of IL-1β and IL-18 by a pyroptotic process involving the cleavage of gasdermin D [1,2,3]. In human monocytes, an alternative pathway activates the NLRP3 inflammasome without the requirement of a second stimulat [4,5]. The second signal necessary to activate NLRP3 inflammasome can be achieved by different stimuli, being the activation of the purinergic P2X receptor 7 (P2X7R) by a concentration of extracellular ATP (eATP) in the mM range among the most common used [6]. We have recently found that activation of P2X7R before priming signal 1, leads to a mitochondrial damage and a later defect on NLRP3 activation [7], highlighting the different pathways associated to P2X7R signaling. Extracellular nucleotides such as ATP are indicative of cellular distress, released in small amounts during apoptosis or in faster, larger amounts during sudden cell death like necrosis or pyroptosis [3,8,9]. In this regard, extracellular nucleotides have been implicated in several inflammatory processes either promoting or reducing the inflammatory response depending on the context [10,11].

Depending the type of nucleotide and its concentration, extracellular nucleotides, and their direct degradation products, bind to different purinergic receptors, including adenosine receptors, metabotropic P2Y and ionotropic P2X receptors, providing a wide variety of immune responses depending on the ligand interaction that goes beyond P2X7R activating the NLRP3 inflammasome [11]. Therefore, it is expected that nucleotides will activate different purinergic receptors expressed in the target cell resulting in different intracellular signaling pathways. This concept led us to hypothesize that initial lower concentration of nucleotides, that do not trigger P2X7R, may regulate NLRP3 inflammasome activity prior its full activation, modulating the overall inflammatory response, beyond the role of P2X7R downmodulating inflammasome response [7]. To test this hypothesis, we employed murine residential peritoneal macrophages (RPMs), a tissue resident macrophage involved in cell clearance during steady-state conditions as well as in inflammatory responses [12,13]. Expression of different purinergic receptors by these cells makes this type of macrophage an excellent model to study the role of nucleotides on NLRP3 inflammasome activation. We found that when RPMs were cultured with nucleotides at μM concentrations that do not trigger P2X7R in the presence of LPS, there was an increase in IL-1β release after NLRP3 activation, and an increase of IL-6 whereas a decrease in TNF-α production in response to the LPS priming. Blockade of purinergic P2Y_2_ receptor (P2Y_2_R) reverted IL-1β levels back to amounts obtained only with LPS, whereas decrease production of TNF-α was due to adenosine receptor activity originated from ATP degradation into adenosine. Our data indicates that nucleotides contribute at different levels to a distinct and unique pro-inflammatory signature, which may be important for future anti-inflammatory therapies.

## 2. Results

### 2.1. ATP and UTP Nucleotides Enhance IL-1β Production by Macrophages

Murine RPMs primed with a combination of LPS with ATP and UTP (NTPs) released higher amounts of IL-1β when NLRP3 inflammasome was activated compared to macrophages only primed with LPS (Figure 1A,B). The increase on IL-1β release was observed using either nigericin or ATP as second signal to induce the activation of the NLRP3 inflammasome (Figure 1B). We then found that the observed increase in IL-1β was the mature p17 form of IL-1β (Figure 1C). This increase was not observed when NLRP3 was activated in THP-1 or bone marrow derived macrophages and cells were priming in the presence of NTPs (Figure 1B).

Individual or combination of ATP and UTP nucleotides increased in a dose-dependent manner the release of IL-1β, with a peaking effect around 20 μM (Figure 1D). Interestingly, no clear synergistic or additive effects were observed when ATP and UTP were combined compared with single nucleotide treatment (Figure 1D), suggesting a common receptor for both nucleotides. Therefore, 20 μM concentration was selected to be used throughout the course of this study unless otherwise mentioned.

Stimulation with different pathogen associated molecular patterns (PAMP) moieties like Poly I:C or Pam3Cys instead of LPS, when combined with nucleotides, resulted in similar upregulation in the release of IL-1β after NLRP3 was activated (Figure 1E). Addition of nucleotides at different time points, before or after LPS, to macrophages showed that the maximal potency was reached when nucleotides were combined with LPS at the same time (Figure 1F). Interestingly, when macrophages were initially incubated with NTPs, then rinsed and incubated with LPS without addition of nucleotides, the release of IL-1β after NLRP3 stimulation remained similar to cells treated only with LPS (Figure 1G), suggesting a requirement for both LPS and nucleotide signals to synergize in order to increase the production of IL-1β.

### 2.2. Nucleotides Increase Il1b Gene Expression, but Do not Affect NLRP3 Priming nor Activation

We then found that NTPs priming did not affect other outcomes of NLRP3 inflammasome activation, and the release of IL-18, pyroptosis (as measured by LDH release), or the percentage of ASC-specking macrophages, were not increased (Figure 2A–C). Furthermore, similar caspase-1 p10 subunit was detected on the supernatants of macrophages primed with NTPs (Figure 2D). Also, gene expression of inflammasome components *Nlrp3*, *Pycard*, *Casp1* or *Il18* was not increased when LPS priming was performed in the presence of NTPs (Figure 2E). NLRP3 protein expression was also not altered when NTPs were added during LPS priming (Figure 2F). However, pro-IL-1β protein increased in macrophages stimulated with LPS and NTPs (Figure 2F). This increase in IL-1β protein, was also confirmed at transcriptional level (Figure 2G), suggesting that the enhanced release of IL-1β after NLRP3 activation could be due to an increase in gene expression.

### 2.3. Nucleotides Induce a Specific Proinflammatory Signature during LPS Priming

Production of inflammasome-independent cytokines like IL-6 and TNF-α was also analyzed in the supernatants of RPMs stimulated for 3 h in the presence of LPS alone or LPS with nucleotides. Nucleotides induced a decrease in TNF-α production, whereas secretion of IL-6 was increased (Figure 3A). Gene expression analysis indicated that TNF-α transcription was also decreased when nucleotides were incubated with LPS (Figure 3B). However, IL-6 gene expression remained unchanged (Figure 3B), indicating that nucleotides did not interfere with IL-6 gene transcription.

Since adenosine receptors have already been described to be involved in decreasing TNF-α mediated inflammatory response [10], we sought to confirm that the observed decrease in TNF-α was due to the conversion of ATP to adenosine. First we confirmed that TNF-α was only decreased when ATP and not UTP was used during LPS priming (Figure 3C), pointing to a degradation of ATP into adenosine as a putative cause. Second, we found that when the inhibitor of adenosine A2A receptor SCH58261 and A2B receptor MRS1754 were used, the decrease in TNF-α production induced by nucleotides was reverted (Figure 3D). Adenosine inhibition did not affected IL-6 release enhanced by nucleotides (not shown). These data indicate that nucleotides induce a specific proinflammatory signature, with some proinflammatory cytokines increased (IL-1β, IL-6) whereas others (TNF-α) are decreased.

### 2.4. Nucleotides Activate P2Y_2_ Purinergic Receptor to Increase IL-1β Production

To further characterize the putative purinergic receptor(s) responsible for the observed increase in IL-1β, we initially performed a qPCR analysis of purinergic receptors expression in RPMs (Figure 4A), finding elevated expression of *P2rx4*, *P2rx7*, and also of *P2ry2*, *P2ry6*, *P2ry12*, *P2ry13* and *P2ry14*. The use of ATP-γS, an ATP analog that cannot be degraded but also an agonist of purinergic receptors, confirmed that ATP, and not a degradation product, was responsible for the increase in IL-1β (Figure 4B).

Knowing that either ATP or UTP were able to induce the increase in IL-1β production (Figure 1D), we first rule out the involvement of the P2X7R (Figure 4C). RPMs express *P2ry2* (Figure 4A), the only purinergic receptor expressed in RPMs able to respond to both nucleotides, so we tested the blockade of this receptor with the specific inhibitor AR-C118925xx. As shown in Figure 4D, RPMs treated with AR-C118925xx prior LPS and NTPs challenge reduced the amount of IL-1β produced after P2X7R activation to levels similar to those induced with only LPS. When NLRP3 was activated by nigericin, AR-C118925xx was also able to decrease IL-1β release when cells were incubated with LPS and nucleotides (not shown). 

Treatment with AR-C118925xx inhibited the increase induced by LPS and ATP alone or LPS and UTP alone (Figure 4E), without inducing cytotoxicity (Figure 4F), indicating that no other purinergic receptor was involved. Similarly, AR-C118925xx was able to slightly decrease IL-6 production enhanced by NTPs (not shown), suggesting that P2Y_2_R and not adenosine receptors could have a role in IL-6 release, as has been shown for other cytokines [14]. We further confirmed the role of P2Y_2_R in this effect when macrophages isolated from animals deficient in this receptor were compared with wild type counterparts and found that nucleotides treatment during LPS priming were not able to affect the release of IL-1β (Figure 4G). P2Y_2_R was also found responsible to the increase in *Il1b* gene expression when nucleotides were present during LPS priming (Figure 4H).

### 2.5. P2Y_2_R-Induced JNK Activation is Responsible for Increased in IL-1β Production

P2Y_2_ receptor is a seven transmembrane receptor that belongs to the G-protein coupled receptor family. Some of these receptors signal through a mechanism that involves activation of phospholipase C (PLC) and intracellular calcium release [15]. However, when RPMs were pretreated with inhibitors of PLC (U73122) or intracellular calcium was blocked with BAPTA-AM or depleted with thapsigargin, the specific increase in IL-1β induced by nucleotides was not reduced (Figure 5A and data not shown). Moreover, cells treated with U73122 and then stimulated with LPS and NTPs for a short period of time (25 min), to avoid cytotoxicity or loss of inhibition, presented higher levels of *Il1b* gene expression when compared to LPS alone (Figure 5B). Similarly, inhibition of the PI3K pathway with LY294002 or wortmannin did not block the nucleotides specific increase of IL-1β (data not shown).

Considering we had observed an increase in *Il1b* expression (Figure 2) we explored common pathways between P2Y_2_R and Toll-like receptors (TLRs) that will result in a transcriptional upregulation. In this regard, mitogen-activated protein kinase (MAPK) activity is found downstream activation of TLRs and purinergic P2Y receptors [16,17,18]. This prompted us to study the effect of nucleotide treatment on the classical ERK1/2, p38 and JNK MAPKs. When specific inhibitors against JNK, ERK1/2 and p38 MAPK (SP600125, U0126 and SB202190 respectively) were used, only the inhibition of JNK reduced the nucleotide-specific increase in IL-1β (Figure 5C), whereas the inhibition of ERK1/2 and p38 MAPKs presented a blockage of IL-1β production independently of the nucleotide treatment. The use of nucleotides alone induced no phosphorylation in JNK (not shown) supporting the idea that nucleotides need to synergize with LPS in order to enhance *Il1b* expression (Figure 1G). The inhibitory effect of MAPK blockers was not due to a cytotoxic effect, since they did not increase LDH release (Figure 5D) and no morphological changes were observed in the cultured macrophages (not shown). 

Interestingly, we then found that nucleotides acting through P2Y_2_R were able to increase the phosphorylation of JNK and ERK when combined with LPS (Figure 5E). However, nucleotides did not affect p38 phosphorylation (Figure 5E). Macrophages from mice deficient in P2Y_2_R confirmed that nucleotides triggering this receptor were important for JNK phosphorylation (Figure 5F). Inhibition of P2Y_2_R with AR-C118925xx resulted in a reduced JNK phosphorylation pattern similar to that obtained with the JNK inhibitor SP600125 (Figure 5G). Interestingly, JNK inhibition also prevented pro-IL-1β enhancement induced by NTPs (Figure 5H). In our system, we also observed that addition of fetal bovine serum to the cell culture, a well-known JNK activator, when cells were stimulated only with LPS, increases basal JNK phosphorylation and production of IL-1β is equal in conditions with or without NTPs (not shown), supporting our results relating JNK activity to IL-1β production.

### 2.6. High Cell Density Disables Nucleotide-Induced IL-1β Production

MAPK signaling is dependent on multiple factors, including stress and survival signals as well as intercellular contact. Since we did not observe any effect on macrophage viability during activation or inhibition of MAPK, we decided to study their signaling on different cell confluences. Macrophages seeded at high cellular concentration (>10^6^/mL) resulted in low differences in IL-1β release between cells primed only with LPS or with LPS and nucleotides (Figure 6A). Only when cells were plated at lower concentration (<0.5 × 10^6^/mL) the enhancing action of NTPs in IL-1β release was observed (Figure 6A). 

We hypothesized that MAPK activity would be different when cells were plated at high density compared to low density to explain the differences. We proceeded to compare MAPK activation in macrophages plated at low cellular density (0.3 × 10^6^/mL) vs. high cellular density (1.2 × 10^6^/mL). As shown in Figure 6B, both MAPK levels and activity were increased when macrophages were cultured at high cell density and no increase in phosphorylation was observed when NTPs were incubated with LPS. This may be indicating that intercellular contacts in high cellular density already induce a signal through JNK that nucleotides cannot boost further up.

To confirm the importance of cell density in IL-1β production, we first changed not only cell concentration in the same plate, but we also changed plating area while maintaining the same amount of cells used at high cell density. For this purpose, we cultured 1.2 × 10^6^ macrophages in 96-well plates (small area, high density) and compared to the same quantity of macrophages cultured in 24-well plates (large area, small density) from the same plastic source brand. Figure 6C shows that nucleotides failed to enhance IL-1β production from macrophages in 96 well plates but were again able to increase IL-1β release when cultured in 24 well plates at low density.

As a final approach to confirm the importance of cell density in IL-1β production, we co-cultured inflammasome competent wild-type macrophages with NLRP3 inflammasome or caspase-1/11 deficient macrophages. Figure 6D shows that increasing cell density of a constant number of wild-type macrophages with the addition of NLRP3-deficient macrophages resulted in an increased IL-1β production independently of nucleotide signaling. This compensation was also observed when caspase-1/11 deficient macrophages were employed (data not shown).

Altogether, our results suggest that macrophages in the presence of PAMPs respond to nucleotides activating JNK through P2Y_2_R engagement, increasing IL-1β levels only in low cell density conditions. When cells are cultured in high density, macrophages produce more IL-1β as a consequence of increased basal JNK activity, rendering nucleotide signaling inefficient.

## 3. Discussion

Macrophages are important immune cells to control the initiation of the inflammatory response due to the expression of a wide array of receptors [19], and therefore are highly sensitive to stimulation with PAMP or DAMP moieties that will induce production of pro-IL-1β cytokine as well as activation of the NLRP3 inflammasome to generate the maturation of this cytokine [20]. Nucleotides control at different levels the inflammasome-related production of IL-1β and IL-18, however there are reports indicating nucleotides could either induce or inhibit their production [21,22]. The idea that nucleotides influence the NLRP3 inflammasome through purinergic receptors other than P2X7 receptor has already been suggested and reviewed [23,24], but this hypothesis require better characterization, which prompted us to analyze in detail the effect of different nucleotides on macrophages. In this study, we describe that when murine peritoneal macrophages were primed with endotoxin in the presence of low nucleotide concentrations (in the range of 2–200 μM), such as ATP and/or UTP, there is an induction of IL-1β production through P2Y_2_R after NLRP3 activation, whereas caspase-1 activation, IL-18 production, ASC speck formation and pyroptosis remained unchanged. This effect is most probably explained by an increase in *Il1b* gene expression induced by P2Y_2_R controlling JNK signaling, and not in changes in NLRP3 activity. On the contrary, when extracellular ATP concentration rises to the mM range and P2X7R activates before macrophage priming with endotoxin, we recently found that the activity of NLRP3 inflammasome is reduced [7].

Extracellular ATP is able to activate a wide range of purinergic receptors in target cells, and its concentration will dictate the type of receptor activated [25]. While low concentrations of ATP activate P2Y receptors and P2X1–6 receptors, higher concentrations are needed to activate P2X7R [25]. P2X7R is linked to different signaling pathways, and its activation could contribute not only to the activation of the NLRP3 inflammasome, but also affects the cellular energy metabolism, host-pathogen interactions and cell death [26].

Interestingly, other inflammasome-independent proinflammatory cytokines, like IL-6 and TNF-α, were increased and decreased respectively, indicating a specific inflammatory response of macrophages when confronted to PAMPs and nucleotides. Decrease of TNF-α has already been described when adenosine receptors are activated together with LPS [10,27], and our study also confirmed adenosine receptors as responsible for TNF-α decrease, probably due to ATP degradation to adenosine. The fact that nucleotides imprint a specific inflammatory signature in macrophages has important consequences for anti-inflammatory treatment therapies in diseases such as inflammatory bowel disease, autoimmune arthritis, cardiovascular diseases, cancer or even obesity [28,29,30,31]. IL-6 is found increased after nucleotide treatment, suggesting that P2Y_2_R could favor the release of cytokines, similarly to the effect of P2Y_2_R found to be able to release MCP-1 without altering the mRNA levels [14].

Our study found P2X4 receptor as highly expressed in mouse peritoneal macrophages, as has been previously reported [32]. However, the relatively high expression of *p2yr2* gene as well as response to low concentrations of ATP or UTP directed us to target P2Y_2_R as the sensor responsible for the observed increase in IL-1β. Nonetheless, we cannot rule out a combined effect of different purinergic receptors affecting the increase of IL-1β, whose effect is reduced if one of the receptor’s signaling, as P2Y_2_R, is absent or blocked. In fact, P2X4 receptor has been implicated in IL-1β release [33,34] and could also modulate the increase of IL-1β when ATP was applied to the macrophages. We found that the inhibition or genetic deficiency of P2Y_2_R restored NLRP3-dependent IL-1β release to levels obtained only with LPS priming. Given that P2Y_2_R signaling affects pro-IL-1β synthesis, the increase in IL-1β production should not be restricted to the activation of NLRP3, but also could affect other inflammasome activation, such as NLRC4. However, the differential presence of P2Y_2_R in different type of macrophages will shape the response to ATP or UTP, since this effect is not present in THP1 or BMDM. This could be due to either a lack of P2Y_2_R receptor or membrane expression or P2Y_2_R might not be similarly coupled to JNK signaling and differently coupled to other pathways. The role of P2Y_2_R is unclear in the inflammatory processes, with paradoxical reports. P2Y_2_R has been involved in cell clearance processes and thus helping maintaining homeostasis [8]. However, P2Y_2_R has also been shown to be involved in pro-inflammatory responses [35,36]. Recently, it has been described that P2Y_2_R is required to induce IL-1β production in irradiated tumor cells by a pannexin-1-dependent mechanism [37], although the downstream P2Y_2_R signaling is unknown, it would suggest a physiological context in which macrophages would show a pro-inflammatory response as indicated in this manuscript. P2Y_2_R, as many G protein-coupled receptors, signals through PI3K, PLC activation and a subsequent intracellular calcium release [14], although the exact mechanism remains unclear [38,39]. However, we have not observed a requirement for PI3K, PLC or intracellular calcium signaling when P2Y_2_R increased IL-1β production, suggesting an alternative pathway of the classical P2Y_2_R activation, as it has been described for endothelial cells [40]. Similarly, PLC-independent mechanisms by which P2Y_2_R can modify cytokine production in macrophages have also been described [14], although the exact mechanism remains unclear.

The observation that *Il1b* gene expression was increased by P2Y_2_R activation prompt us to test other pathways that were also downstream purinergic receptors and induced *Il1b* gene transcription, and MAPK have been recently found downstream purinergic receptors in human monocytes [41,42]. We found that inhibition of JNK MAPK resulted in a decrease in nucleotide-induced IL-1β, whereas inhibition of the other two classical MAPK, ERK1/2 and p38, resulted in a general IL-1β decrease, but did not affect P2Y_2_R-increased IL-1β release. The importance of JNK activity in IL-1β production by macrophages has already been described for the activation of the inflammasome by calcium crystals [43] or palmitate [44]. Increased NLRP3 activation by ATP or UTP has also been described; however, P2Y_2_R was not responsible for this activation [45]. Furthermore, JNK has been implicated in the direct activation of NLRP3 inflammasome by phosphorylating its PYD domain and favoring ASC engagement [46]. Our results confirm that P2Y_2_R was not affecting the activation of NLRP3, but enhanced *Il1b* transcription by JNK activation, and this model is not incompatible with a direct NLRP3 phosphorylation.

We also describe how nucleotides influence macrophage response depending on cell density. A common approach used in experiments to assess inflammasome activation is to set cell cultures at full confluence. However, macrophages in vivo are present more dispersed in steady state conditions or even during pathological settings, with some exceptions as parasite infections, foreign object presence, granulomas and some tumor locations, where they aggregate at high concentration. It is therefore of interest to analyze the function of macrophages at low densities. We have found that IL-1β production can be enhanced by nucleotides only when macrophages are cultured at low concentration. This effect correlates with an increased JNK activation in cells at high concentration. Signaling through JNK (originally named stress activated protein kinase) in macrophages is linked to inflammation [47]. High cell density cultures provide neighboring cells with important interactions that resulted in an important increase in MAPK activity (and probably other pathways) and possible more stable signals sustained in time than the activation of P2Y_2_R. As a consequence, much higher IL-1β production is obtained from high density cultures, even when high density is reached supplementing NLRP3-deficient macrophages to low density wild type macrophages.

In summary, future studies to further examine the role of extracellular nucleotides and P2Y_2_R signaling during in vivo inflammatory conditions, as well as their potential as novel receptors to treat inflammation are warranted.

## 4. Materials and Methods

### 4.1. Reagents

ATP, UTP, adenosine 5′-o-(3-thiotriphosphate) tetralithium salt (ATP-γS), and nigericin were purchased from Sigma-Aldrich (St. Louis, MO, USA). AR-C118925xx, SP600125, SB202190, U0126, U73122, SCH58261, MRS1754, BAPTA-AM, thapsigargin, wortmannin were purchased from Tocris Bioscience (Bio-techne, Bristol, UK).

### 4.2. Animals

*C57BL/6* (wild-type) mice were purchased from Harlan Laboratories (Indianapolis, IN, USA) and bred in the local animal facility. NLRP3-, Caspase-1/11- and P2X7R-deficient (*Nlrp3*^−/−^, *Casp1/11*^−/−^ and *P2rx7*^−/−^ respectively) [48,49] mice in *C57BL/6* background were bred in our facilities. For all experiments, mice aged 8–24 weeks were used, in accordance with the University Hospital Virgen Arrixaca animal experimentation guidelines, and the Spanish national (RD 1201/2005 and Law 32/2007) and European Union (86/609/EEC and 2010/63/EU) legislation. According to the cited legislation, local ethics committee review or approval is not needed, because the mice were killed by CO_2_ inhalation and used to obtain peritoneal lavage or tissues. No procedure was undertaken to live animals that compromised animal welfare.

### 4.3. Isolation and Culture of Macrophages

Resident peritoneal macrophages (RPMs) were isolated from resting C57/BL6 mice peritoneal cavity previously euthanized with CO_2_ by lavage using cold PBS with 2 mM EDTA. Macrophages were further enriched by magnetic depletion of CD19^+^ and CD5^+^ cells using magnetic microbeads (Miltenyi Biotech, Bergisch Gladbach, Germany). To confirm macrophage enrichment, initial tests of samples before and after magnetic bead purification were analyzed by flow cytometry for expression of F4/80 (antiF4/80-alexa488, clone BM8, BioLegend, San Diego, CA, USA), MHC-II (anti-MHC-PE conjugated, clone M5/114.15.2, eBiosciences, San Diego, CA, USA) or CD19 (PE-conjugated, clone eBio1D3, eBiosciences) and analyzed in a FACSCanto cytometer (BD Biosciences, San Diego, CA, USA). Cells were plated at 0.3 × 10^6^ cell/mL (low density) or 1.2 × 10^6^ cell/mL (high density) for 1 h in RPMI-1640 media (Life Technologies, Carlsbad, CA, USA) with 10 mM HEPES and 2 mM L-glutamine (BioWhittaker—Lonza, Basel, Switzerland) (supplemented RPMI) and complemented with 5% fetal bovine serum (FBS, Life Technologies). Plate wells were then rinsed with pre-warmed PBS to remove non-adherent cells and further enrich the macrophage culture. Cells were then cultured for a minimum of 2 h in supplemented RPMI with 0.5% endotoxin free, sterile filtered bovine serum albumin (Sigma). Bone marrow derived macrophages (BMDM) were differentiated for 7 days in the presence of L-cell media as already described [50]. Differentiation of THP-1 cells was performed in RPMI media supplemented with 10% FBS and 0.2 μM PMA for 4 h, then media was replaced with fresh media with FBS without PMA and cells were incubated overnight. Cells were then rinsed and media without FBS was added for cell stimulation.

Cells were stimulated for 3 h with either 200 ng/mL of ultrapure LPS from *E. coli* 0111:B (InvivoGen, San Diego, CA, USA), 2 μg/mL of Pam3CSK4 (InvivoGen) or 20 μg/mL of Poly I:C (InvivoGen) in the presence or absence of 20 μM of ATP and/or 20 μM of UTP, unless otherwise indicated. Supernatants from this initial priming step were recovered when needed for cytokine detection, and plates were rinsed with PBS and then with physiological buffer (147 NaCl, 10 HEPES, 13 D-glucose, 2 KCl, 2 CaCl_2_, and 1 MgCl_2_; pH 7.4, all in mM concentration). Finally, cells were stimulated with physiological buffer containing or not 3 mM of ATP or 5 μM nigericin, for 24 min, and then supernatants were recovered, cleared and stored at −80 °C.

### 4.4. LDH Determination

Pyroptosis was analyzed by measurement of released lactate dehydrogenase (LDH) in the supernatants using the Cytotoxicity Detection kit (Roche, Barcelona, Spain) following the manufacturer’s instructions, and expressed as percentage of total cell LDH content, using samples from cells lysed in 1% Triton X-100 buffer.

### 4.5. ELISA

Supernatants from cultured macrophages in duplicate wells were cleared at 500× *g* to remove any remaining cell. IL-1β was analyzed with Affimetrix’ Ready-Set-Go (BioLegend) ELISA kit. TNF-α, IL-6 and IL-18 were analyzed by Quantikine ELISA (R&D, Biotechne, Minneapolis, MN, USA).

### 4.6. Microscopy

RPMs were seeded at the desired concentration (0.4 × 10^6^ or 1.2 × 10^6^ cell/mL) onto coverslips with RPMI media with 5% FBS. After being activated with 5 μM nigericin, 3 mM ATP, cells were fixed in 2% paraformaldehyde. Cells were blocked with autologous serum and stained with primary anti-ASC (HASC-71, BioLegend) and secondary donkey anti-mouse AlexaFluor488 (Molecular Probes, Thermo Fisher Scientific, Waltham, MA, USA) and mounted on slides with DAPI-containing mounting medium (Prolong diamond antifade, Life Technologies). Images were acquired with an Eclipse *Ti* microscope (Nikon, Tokyo, Japan) equipped with a 10× (numerical aperture, 0.30) or 20× S Plan Fluor objective (numerical aperture, 0.45) and a digital Sight DS-QiMc camera (Nikon) and 387 nm/447 nm and 482 nm/536 nm filter sets (Semrock, Lake Forest, IL, USA).

### 4.7. Quantitative Reverse Transcriptase-PCR Analysis

mRNA was obtained using the RNeasy Mini kit (Qiagen, Venlo, The Netherlands) as per manufacturer instructions. Quantitative PCR was performed using SYBR Premix ExTaq (Takara, Göteborg, Sweden). Specific primers were purchased from Qiagen (QuantiTech Primer Assays). For each primer set, the efficiency was >95%, and a single product was obtained on melt curve analysis. The presented relative gene expression levels were calculated using the 2^ΔΔCt^ method normalizing to the endogenous *Hprt1* expression levels, as a house keeping control, for each treatment, and the fold increase in expression was relative to the smallest expression level or to the control basal levels.

### 4.8. Western Blot Analysis

Cells were plated at 0.3 × 10^6^ cell/mL (low density) or 1.2 × 10^6^ cell/mL (high density) were rinsed with cold PBS and lysed in 1% NP40 buffer supplemented with protease inhibitor cocktail (Sigma-Aldrich) and phosphatase inhibitor (PhosSTOP, Roche). Cell lysates and supernatants were resolved in 12% acrylamide SDS-PAGE gels and blotted into a PVDF membrane for mIL-1β (H-153, Santa Cruz, Dallas, TX, USA), NLRP3 (Cryo-2 AG-20B-0014, Adipogen, Liestal Switzerland) or anti MAPK antibodies anti-Phospho JNK (Thr183,Tyr 185, Cat. 9251S), anti-JNK (Cat. 9252S), anti-Phospho-p44/42 (phospho-Erk1/2) (Thr202, Tyr 204, Cat. 4377), anti-p44/42 (Erk1/2), anti-phospho p38 (Thr 180/Tyr 185, Cat. 9211) or anti-p38 (Cat. 9212) (all from Cell Signaling Technology, Danvers, MA, USA). Primary antibody incubation was performed overnight in 3% bovine serum albumin (Sigma-Aldrich) or 5% *w/v* Difco skim milk (BD Biosciences). Primary antibodies were revealed using the corresponding secondary anti-mouse, or anti-rabbit IgG-peroxidase horseradish linked (GE-Healthcare, Munich, Germany). Analysis of protein bands was performed using Image Lab software (Bio-Rad Laboratories, Hercules, CA, USA), and values were normalized to β-actin.

### 4.9. Statistical Analysis

Data is shown with +SEM or +SD as indicated. For some experiments (mainly ELISAs) data was normalized dividing tests values by that obtained by control (i.e., LPS only, or no inhibitor control), thus, control test was given value “1” and the tests results are proportional to that control. In other experiments, due to large differences in the potency of the response of the cells in the different repeated experiments, we show a representative experiment of the replicates. Statistical analysis was performed using Prism software (GraphPad Inc., La Jolla, CA, USA) by testing two-way ANOVA with Sidak’s multiple comparison test for multiple comparisons or by multiple t-test between 2 groups.

## Figures and Tables

**Figure 1 ijms-21-04686-f001:**
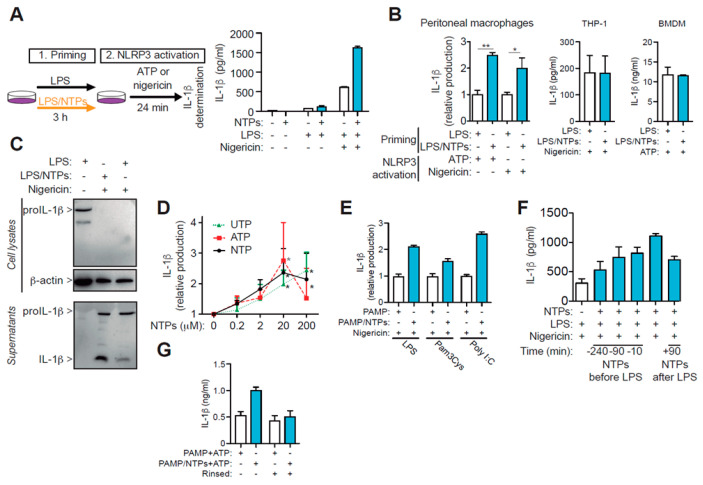
Nucleotides increase LPS-primed IL-1β production. (**A**) Schematic representation of the protocol employed to activate the NLRP3 inflammasome (left). Representative analysis of IL-1β produced by peritoneal macrophages (RPM) plated in duplicates at 0.3 × 10^6^ cell/mL and primed with LPS, in the absence (white) or presence (blue) of 100 μM ATP and UTP (NTPs) for 3h and then the NLRP3 inflammasome was activated with nigericin; bars indicate +SD. (**B**) IL-1β production from RPMs, mouse bone marrow derived macrophages (BMDM) or THP-1 cells primed with LPS in the absence (white bars) or presence (blue bars) of nucleotides when NLRP3 was activated with ATP or nigericin as indicated; average +SEM of 12 and eight experiments, respectively, is shown for RPMs, one representative experiment out of *n* = 3 different experiments for BMDM and *n* = 2 for THP-1. (**C**) Representative analysis by Western blot of cell lysate and supernatants of macrophages primed with LPS only (left and right lanes) or LPS with NTPs (middle lane) and then activated with nigericin as indicated. (**D**) Dose-response effect of nucleotides priming with LPS on IL-1β production as indicated in (**A**); data is normalized to IL-1β production of RPMs primed with LPS absence of nucleotides and shows the average of three experiments +SEM. (**E**) IL-1β production in supernatants from RPMs primed with the indicated PAMP in the presence (blue bars) or absence (white bars) of NTPs (20 μM UTP/ATP) for 3 h and following NLRP3 activation with nigericin. One experiment +SD out of three is shown. (**F**) Kinetic analysis on IL-1β production when NTPs were added prior (negative values) at the same time or after (positive values) cells were primed with LPS and after 3 h NLRP3 was activated with nigericin; a representative experiment out of 2–4 is shown, bars indicate +SD. (**G**) Resting RPMs were primed with NTPs (20 μM UTP/ATP) or not for 10 min. before adding LPS, then cells rinsed (right pair of columns) or not (left pair of columns). LPS was then added for 3h and NLPR3 inflammasome was activated using 3 mM of ATP for 24 min. Supernatants from both conditions were compared for IL-1β production. A representative experiment of three is shown with SD. * = *p* < 0.05 or ** = *p* < 0.01 significance.

**Figure 2 ijms-21-04686-f002:**
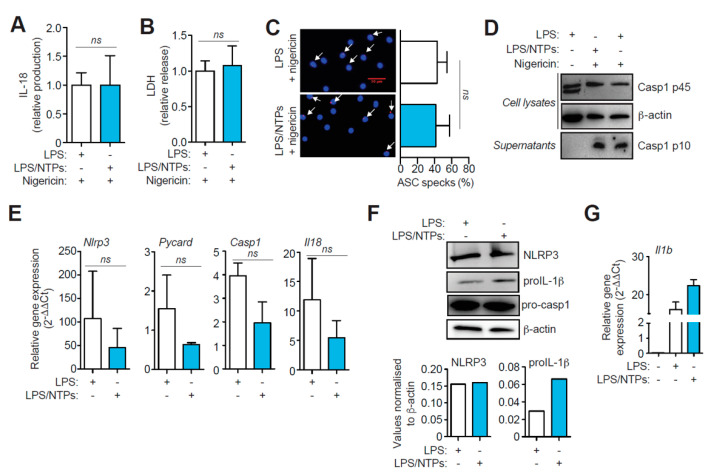
Nucleotides prime *Il1b* production but do not affect NLRP3 inflammasome activation. (**A**) ELISA of IL-18 produced by peritoneal macrophages (RPMs) primed with LPS in the absence or presence of nucleotides (NTPs: 20 μM UTP/ATP) and then activated with nigericin. Average of four experiments is shown +SEM, corresponding to an average of 87.8 pg/mL of IL-18 in LPS + nigericin condition. (**B**) LDH release analysis from RPMs treated as in (A). Normalized average +SEM of 16 experiments is shown. (**C**) RPMs treated as in (A) and stained for ASC (red) and nuclei (DAPI, blue) (left panels). White arrows indicate ASC specking macrophages. Red bar indicates scale (30 μm). Quantification of the percentage of ASC specking macrophages of three fields from duplicates experiments (right panels); graph represents the average of five experiments +SEM. (**D**) Representative analysis by Western blot of cell lysate and supernatants of macrophages primed with LPS only (left and right lanes) or LPS with NTPs (middle lane) and then activated with nigericin as indicated; β-actin loading control is the same shown in Figure 1C, as there are the same blots shown in that figure but revealed with anti-caspase-1 antibody. (**E**) qPCR of the indicated genes after 3 h with LPS (white bars) or LPS+NTPs (blue bars) to *Hprt1* housekeeping gene expression and represented as 2^-ΔΔCt^ values; average of 3 experiments is shown +SD. (**F**) Western blot for NLRP3, IL-1β, pro-caspase-1 and β-actin from cells treated as in (**D**) (top panel); normalized expression of NLRP3 and IL-1β to β-actin expression (bottom panel). (**G**) qPCR analysis of *Il1b* gene from cells treated as in (**D**). A representative experiment with duplicates out of 3 experiments performed is shown +SD. *ns* = not significative.

**Figure 3 ijms-21-04686-f003:**
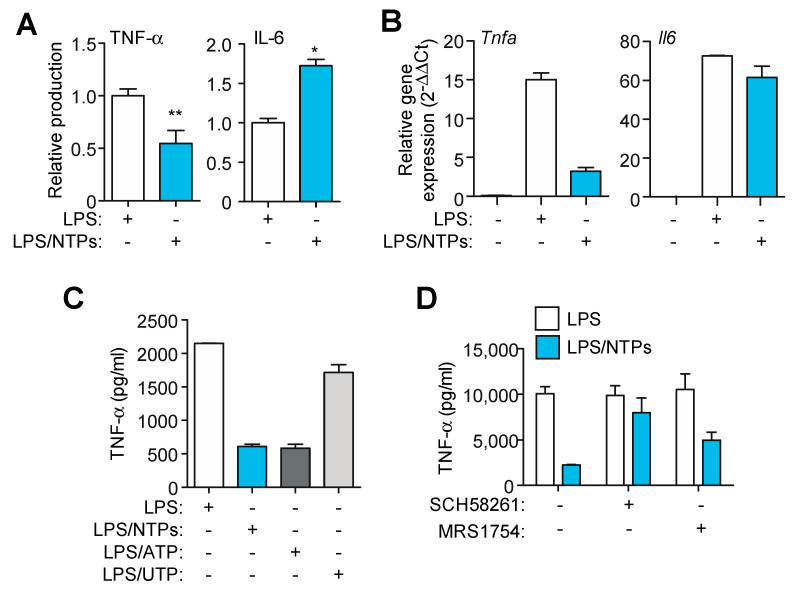
Nucleotides decrease TNF-α while increase IL-6 production. (**A**) ELISA for TNF-α and IL-6 in supernatants from peritoneal macrophages (RPMs) treated with LPS for 3 h in the presence (blue bars) or absence (white bars) of nucleotides (NTPs: 20 μM UTP/ATP) without activating the NLRP3 inflammasome. Normalized data shows average +SEM from eleven (TNF-α) and seven (IL-6) experiments. (**B**) qPCR analysis of TNF-α and IL-6 from RPMs treated as in (A); a representative experiment of three is shown +SD. (**C**) ELISA for TNF-α from cells treated as in (A), but combining nucleotides or separating them as indicated, one representative experiment out of three is shown +SD. (**D**) Supernatants from RPMs pretreated with the indicated inhibitors before being activated were analyzed for TNF-α; a representative experiment out of two is shown +SD. * = *p* < 0.05 or ** = *p* < 0.01 significance.

**Figure 4 ijms-21-04686-f004:**
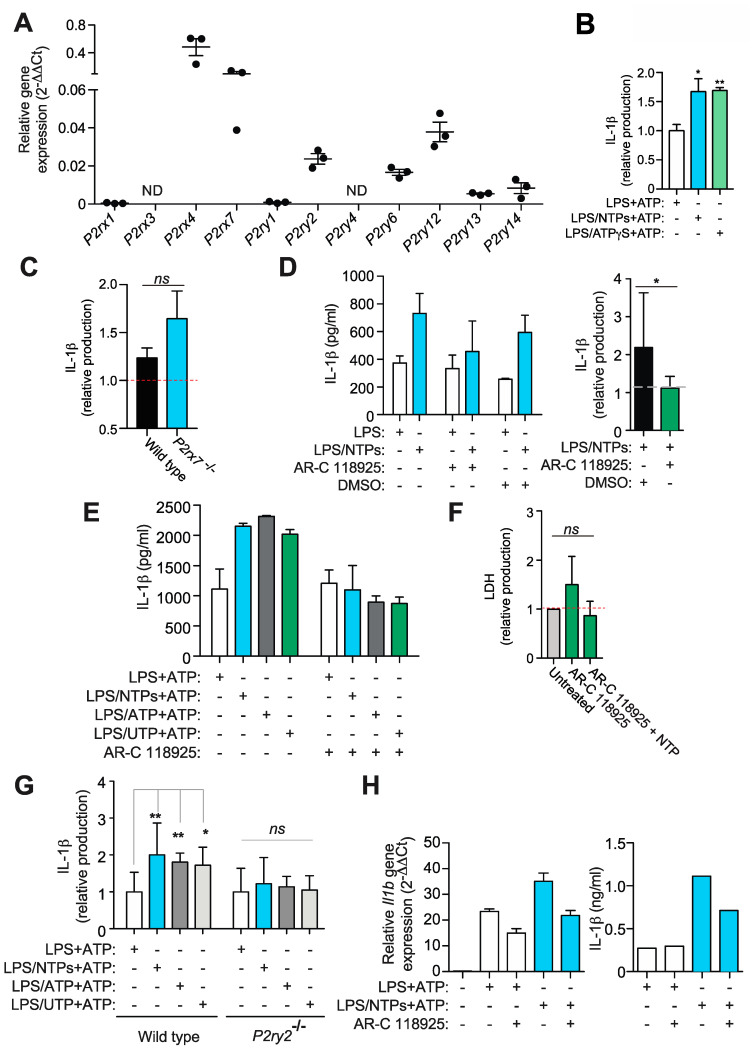
Nucleotides signal through P2Y_2_R to increase IL-1β production. (**A**) Expression by qPCR of purinergic receptors from resting peritoneal macrophages (RPMs) from three experiments +SEM. (**B**) RPMs were treated with NTPs (blue bar, 20 μM UTP/ATP) or ATP-γS (green bar) for 10 min before adding LPS for 3 h and then NLPR3 inflammasome was activated using 3 mM ATP for 24 min. Supernatants collected were analyzed for IL-1β production. Average of three experiments +SEM is shown. (**C**) Normalized IL-1β release from wild type or *P2rx7*^−/−^ RPM primed with LPS+NTPs and then activated with nigericin; relative values to RPM primed with LPS alone and then treated with nigericin (red dotted line). Average of three experiments +SEM is shown. (**D**) A representative experiment of five showing IL-1β production +SD by RPMs pretreated with or without the P2Y_2_R inhibitor AR-C118925xx or DMSO vehicle as indicated before priming with LPS in the presence (blue bars) or absence (white bars) of nucleotides (NTPs: 20 μM UTP/ATP) and then activated for NLRP3 with 3 mM ATP (left panel). Average of seven experiments is shown on graph on the right side +SEM. Dashed line indicates normalized IL-1β production value to LPS without nucleotides. (**E**) IL-1β ELISA from RPMs supernatants primed with LPS alone (white bars) or together NTP (blue bars, 20 μM UTP/ATP), ATP (grey bars) or UTP (green bars) before NLRP3 activation with 3 mM ATP. One experiment out of three is shown +SD. (**F**) Extracellular LDH as a marker of cell death from RPMs treated with AR-C118925xx in the presence or absence of NTPs. (**G**) IL-1β ELISA analysis of supernatants from wild type or *P2ry2*^−/−^ RPMs stimulated as in (**E**); an average +SEM from samples of seven experiments is shown. Data was normalized to LPS alone. (**H**) Supernatants from RPMs treated as in (**D**) were analyzed for IL-1β by ELISA and compared with the expression of *Il1b* gene by qPCR from the same RPMs cell lysates obtained before NLRP3 activation; a representative experiment of 2 is shown +SD. * = *p* < 0.05 or ** = *p* < 0.01 significance; *ns* = not significative; ND: Not detected.

**Figure 5 ijms-21-04686-f005:**
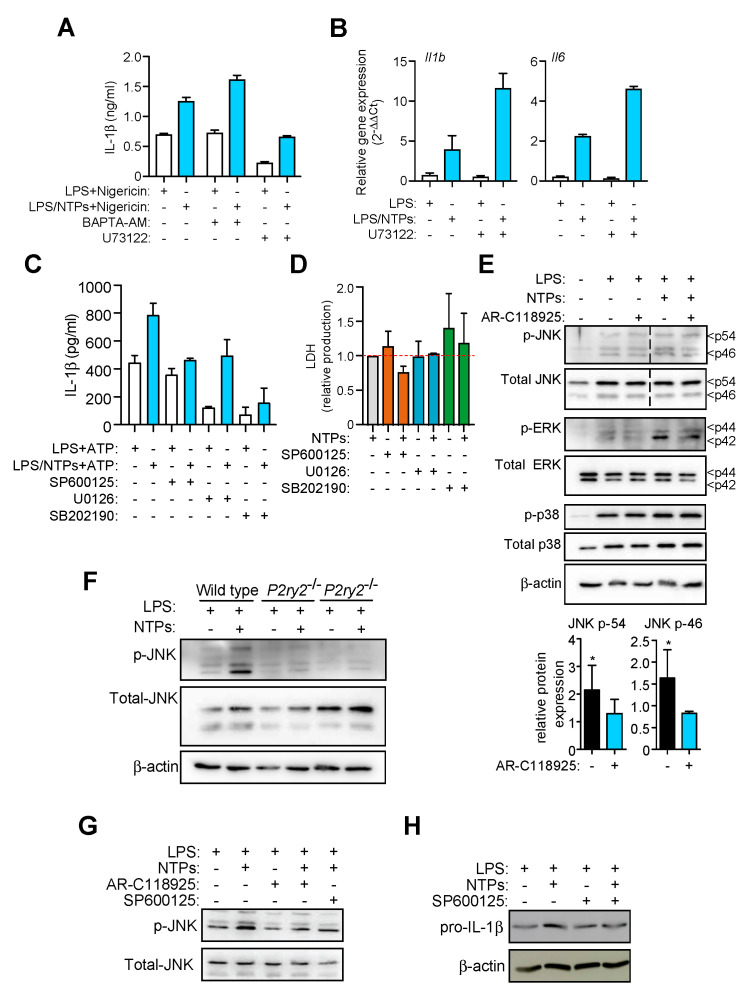
Nucleotides induce an increase in MAPK JNK activity to enhance IL-1β production. (**A**) IL-1β ELISA in supernatants from resting peritoneal macrophages (RPMs) pretreated with BAPTA-AM at 13 μM or the PLC inhibitor U73122 at 2.5 μM for 20 min prior LPS priming with (blue bars) or without (white bars) NTPs (20 μM UTP/ATP) and then NLRP3 was activated with nigericin. A representative experiment of five is shown with +SD. (**B**) Expression by qPCR analysis of *Il1b* and *Il6* genes from mRNA extracted from RPMs preincubated or not with U73122 at 5 μM for 20 min and then primed with LPS with or without NTPs (20 μM UTP/ATP) for 2 h. Graph shows one experiment of two represented as relative gene expression normalized to *Hprt1* expression +SD. (**C**) IL-1β ELISA from peritoneal macrophages (RPMs) supernatants pretreated or not with the indicated JNK, ERK1/2, or p38 (SP600125, U0126, or SB202190 respectively) inhibitor for 20 min prior LPS priming with (blue bars) or without (white bars) nucleotides (NTPs: 20 μM UTP/ATP) for 3 h following NLRP3 activation with 3 mM ATP. One representative experiment out of 3 is shown +SD. (**D**) Extracellular LDH as a marker of cell death from RPMs treated with SP600125, U0126 or SB202190, in the presence or absence of NTPs. Average of four (SP600125) or two (U0126, SB202190) experiments +SEM is shown. (**E**) Lysates from RPMs pretreated with AR-C118925xx inhibitor and stimulated for 25 min with LPS with NTPs (20 μM UTP/ATP) were analyzed for MAPK levels by western blot (top). Dashed line indicates lanes were eliminated from the image to compose the final picture. One experiment out of 3 is shown. Quantification of band intensity of four experiments +SEM of phospho-JNK p-54 and p-46 normalized to total JNK is shown (bottom). (**F**) RPMs cell lysates from wild-type or *P2ry2*^−/−^ mice were analyzed by western blot for the indicated proteins; one experiment out of 2 is shown. (**G**) Lysates from RPMs pretreated with or without AR-C118925xx or SP600125 inhibitor for 20 min and then stimulated for 25 min with LPS or LPS+NTPs (20 μM UTP/ATP) were analyzed for JNK protein levels by western blot. (**H**) Lysates from RPMs pretreated with or without SP600125 inhibitor for 20 min and then stimulated for 2 h with LPS or LPS+NTPs (20 μM UTP/ATP) were analyzed for pro-IL-1β and β-actin protein levels by western blot. * = *p* < 0.05 significance.

**Figure 6 ijms-21-04686-f006:**
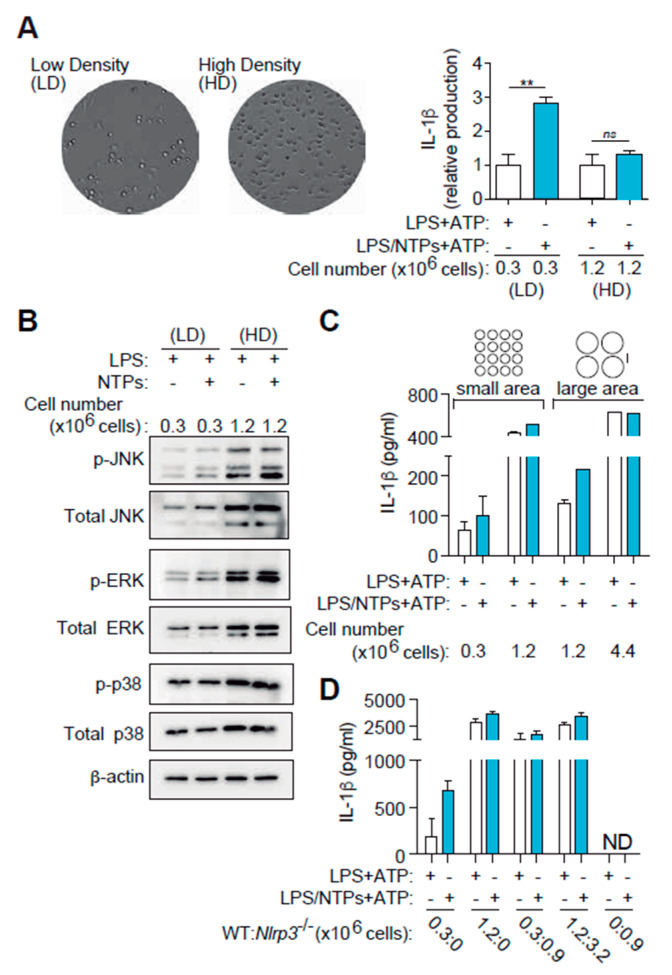
High cell density induces and increase in basal MAPK signaling overriding nucleotide effect on IL-1β production. (**A**) Differential interference contrast (DIC) images from cells plated at low density (LD, left side) or high density (HD, right) on a p24 well plate and stimulated for 2 h with LPS. Graph on the right shows the average of seven experiments +SEM of IL-1β detected by ELISA in supernatants from peritoneal macrophages (RPMs) plated at the indicated concentrations and LPS-primed with (blue bars) or without (white bars) NTPs (20 μM UTP/ATP) for 3 h and then NLRP3 was activated with 3 mM ATP (right plot). Data is normalized to LPS values (white bars). Average of 9 experiments +SEM is shown. (**B**) Western blot for MAPK in RPMs lysates plated at LD or HD and stimulated for 25 min with LPS alone or with NTPs as indicated. One experiment out of two is shown. (**C**) IL-1β ELISA in RPMs supernatants plated at LD or HD in p96 plate (small area) or p24 plate wells (large area) and stimulated as in (**A**); a representative experiment out of three is shown +SD. (**D**) RPMs from wild-type mice were plated in wells of the same area size at LD or HD. RPMs from NLRP3-deficient mice were used to complement confluence as in HD (third column pair) or in saturated conditions (fourth column pair). An experiment out of 3 is shown +SD. ** = *p* < 0.01 significance; *ns*: non-significant; ND: Not detected.

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
