# Peer review of "Signaling Through Purinergic Receptor P2Y2 Enhances Macrophage IL-1β Production"

_ijms, 2020, doi:10.3390/ijms21134686_

Round 1

Reviewer 1 Report

The authors here describe the role of IL-1ß signaling in macrophages through P2Y2. This builds on prior research from this group elucidating the role of P2 type receptor signaling in NLRP3 signaling in infection and inflammation. This authors describe the following:

  • NTPs can enhance LPS-primed IL-1ß secretion in macrophages
  • NTPs can prime IL-1ß but without impact on NLRP3 inflammasome activation
  • NTPs result in TNF-α is decreased while IL-6 is increased with LPS-priming
  • implication of the role of P2Y2 in IL-1ß secretion
  • NTPs result in MARK JNK signaling in IL-1ß secretion that is dependent on cell density

This is a rich body of work that raises interesting discussion regarding the role of P2Y2 in IL-1ß signaling in macrophages. Major concerns for this manuscript are raised by the lack of statistical analysis in many of the figures and the fact that data are largely represented as a representative experiment of 2, 3, or 6. This raises concerns for generalizability of any of the data points; coupled to the lack of statistical power raises question for which are real observations that would be reproducible. Statistical analysis on the full dataset would add greater strength to the manuscript and help to parse those phenomena that are worth investigating. Finally, it is not clear why the authors chose to pursue P2Y2 and not P2X4 and/or P2X7; line 265 "we first rule out the involvement of the P2XR7" - why especially when expression by qPCR is highest in RPMs (figure 4A). Minor: throughout the manuscript, there are some grammatical errors and the P2X7R is written as P2 x 7R in multiple places.

Reviewer 2 Report

The authors examined how UTP and ATP enhance IL-1b production during lipopolysaccharide (LPS)-induced murine resident peritoneal macrophage activation before NLRP3 inflammasome activation. They propose the mechanism involves P2Y2 receptor activation due to JNK activation. Their findings are timely. There are some points that deserve authors attention to improve their work.

Comments:

- Page 1, line 23. The authors stated: “The control of the maturation and release of the proinflammatory cytokine interleukin (IL)-1β 24 and IL-18 in macrophages by the NLRP3 inflammasome complex is a tightly controlled two-step 25 process: an initial signal that primes the production of the immature form of the cytokines and 26 NLRP3, and a second signal that will induce activation of the NLRP3 inflammasome complex …”. The authors should consider rewrite this sentence considering that in humans it is not necessary “two steps”.

- Authors should standardize the spelling of purinergic receptors to the correct form: P2X7 and P2Y2 (2 is subscribed).  Please, review throughout the text.

- My major criticism is the large number of experiments representative of two or even three experiments. Authors have to repeat their experiments and represent them as means + SD of at least three experiments with the necessary statistical analysis.

Reviewer 3 Report

This manuscript describes that UTP and ATP enhance IL-1beta production in LPS-induced inflammation. The authors show that P2Y2 receptors and JNK signaling are involved, but that the production of other cytokines such as TNF-alpha is decreased by nucleotides. The authors conclude that nucleotides shape the response of macrophages to obtain during inflammatory processes.  This manuscript is interesting, but it requires some modifications. Unless already available, additional data are not essential, particularly in light of the difficult situation with the global pandemic.

Main problems:

  1. The abstract is not very clear. For example, the term “before NLRP3 inflammasome activation” needs to be better clarified.
  2. The work presented here is interesting but not entirely novel. For example, recent studies by Lee at al. (PMID: 30247270) and Sueyoshi et al. (PMID: 30787105) have shown that ATP and UTP contribute to IL-1 production and via P2Y2 receptors and that other ATP receptors also contribute to the inflammatory response of monocytes and macrophages. The authors should discuss how their results conform of clash with these previously published findings.
  3. The involvement of P2Y11 receptors should be addressed with additional experiments or in the discussion. Mice lack this ATP receptor, which raises the question how the cytokine production responses of mouse and human cells differ.
  4. Did the authors also study expression patterns of P2 receptors in human cells?
  5. P2X4 was the most highly expressed purinergic receptor. Did the authors study the involvement of this receptor in their experimental settings?

Minor issues:

  1. The text is not written very well and it is sometimes hard to follow what the authors want to say. It is highly recommended that the authors seek the help of a native English speaker to modify the text for clarity and proper grammar.
  2. The use of a mathematical formula for the naming of P2X receptors is not correctly. Instead of P2 7R, for example, a more standard abbreviation such as “P2X7”or “P2X7R” should be used. This may not be the authors fault but is typically an error that type setters seem to make.
  3. In Fig. 2C, no red staining can be seen. There seems to be a problem with color reproduction or a similar issue that must be addressed.

Round 2

Reviewer 1 Report

Thanks to the authors. This statement continues to raise concern:

"The lack of statistical analysis is mainly due to differences in the potency of response of the cells in the different repeated experiments. However, same results (with different potency) were obtained in those sets referred as "representative experiment", leading us to state whether the comparison was different or not in the indicated situations." The data need to be normalized across experiments to convey significance; otherwise one representative experiment does not appropriately capture the rigor and reproducibility required.

This is indicated for Figure 1B, 2G, 3B, 3D, 4D, 4E, 4H, 5A-C, 6C-D,